# Curriculum Learning with Infant Egocentric Videos

**Saber Sheybani**
Department of Intelligent Systems Engineering
Indiana University Bloomington
sheybani@iu.edu

**Himanshu Hansaria**
Department of Computer Science
Indiana University Bloomington
hhansar@iu.edu

**Justin N. Wood**
Department of Informatics
Indiana University Bloomington
woodjn@iu.edu

**Linda B. Smith**
Department of Psychological and Brain Sciences
Indiana University Bloomington
smith4@iu.edu

**Zoran Tiganj**
Department of Computer Science
Indiana University Bloomington
ztiganj@iu.edu

## Abstract

Infants possess a remarkable ability to rapidly learn and process visual inputs. As an infant's mobility increases, so does the variety and dynamics of their visual inputs. Is this change in the properties of the visual inputs beneficial or even critical for the proper development of the visual system? To address this question, we used video recordings from infants wearing head-mounted cameras to train a variety of self-supervised learning models. Critically, we separated the infant data by age group and evaluated the importance of training with a curriculum aligned with developmental order. We found that initiating learning with the data from the youngest age group provided the strongest learning signal and led to the best learning outcomes in terms of downstream task performance. We then showed that the benefits of the data from the youngest age group are due to the slowness and simplicity of the visual experience. The results provide strong empirical evidence for the importance of the properties of the early infant experience and developmental progression in training. More broadly, our approach and findings take a noteworthy step towards reverse engineering the learning mechanisms in newborn brains using image-computable models from artificial intelligence.

## 1 Introduction

A common, implicit assumption in the machine-learning literature is that at the massive scale of daily life, the world presents the same visual statistics to all perceivers. Many researchers therefore believe that it will be possible to mimic biological learning through incremental statistical learning performed over large datasets. However, this assumption may be incorrect because the input received by the visual system depends on the momentary spatial relation of the eyes to the world and is thus dependent on the behavior and goals of the perceiver. Over the first year of life, infants' motor abilities, behaviors, and interests change substantially [Lee and Galloway, 2012, Largo and Kakebeeke, 1994], producing gradually changing datasets for learning. A growing literature on infant egocentric vision also indicates that the semantic content of visual information (faces, hands, objects, scenes) changes systematically with development over the first year post birth as a product of changing motor skills and interests [Smith et al., 2018, Jayaraman et al., 2017, Kretch et al., 2014, Jayaraman et al., 2015]. Experimental studies provide converging evidence of systematic changes in the types of semantic content that infants prefer – and choose – to look at [Spelke, 1985, Frank et al., 2014, 2009, Kwon

et al., 2016]. In brief, the human visual system develops in context of an ordered set of visual experiences, a curriculum. Does this curriculum matter? Would a randomly ordered aggregation of all the data be just as good? From a machine-learning perspective, are there benefits to "developmentally structured" datasets?

The study reported here is motivated by three goals: (1) to determine if the natural order of the developmental curriculum has advantages for machine learning; (2) to determine properties of the ordered data sets that lead to this advantage; and (3) to explore the implications of this self-generated curriculum for building more robust artificial visual systems that can learn continuously from self-supervised training signals. We used the Homeview dataset[1] of egocentric videos recorded by head-mounted cameras worn by infants from 2-12 months of age as they went about their daily lives at home and without experimenters present. This dataset provides unique insight into infant visual experience at different stages of development. The entire data set contains videos captured by very young infants with naturally limited postural control (2 month old) to older infants (12 month olds) who crawl or walk and instrumentally use objects. To evaluate the impact of these changing experiences on visual learning, we performed computational experiments by selecting and ordering training videos from the corpus collected by infants at various ages. These videos were then used to train a variety of state-of-the-art self-supervised learning (SSL) models. We found that one key developing property in the input was the duration of self-similar inputs, with the youngest infant experiences characterized by very slow change. This finding underscores the integral role of curriculum learning in infant development and provides insights relevant to the continued evolution of self-supervised learning systems.

## 2    Related Work

### 2.1    Learning from child egocentric videos

There is increased recognition of the importance of the structure of the training data in AI and use of egocentric infant and child images and videos as inputs to deep neural networks [Bambach et al., 2017, 2016, 2018, Orhan et al., 2020, Zhuang et al., 2021]. One key finding is that these egocentric training data are capable of furnishing powerful visual representations [Orhan et al., 2020, Orhan and Lake, 2023] and can lead to superior learning over adult egocentric videos [Bambach et al., 2018].

In [Bambach et al., 2018], the authors fine-tuned a deep convolutional neural network (CNN) using egocentric images recorded by toddlers and adults who were in the same environment during data collection and found that images recorded by toddlers led to better subsequent classification of objects. Analyses of the differences in the training set suggested two statistical properties in the infant corpus that may be critical to learning: (1) the distribution of similarities among images of the same object was extremely skewed such that most images were variations of each other, but the overall range (the long tail) of variability in the infant images was much greater than that in the adult images; and (2) individual objects were much larger and foregrounded in the images – and thus easier to segment from the background than were the same objects in the adult images. This result emerges because active toddlers visually attend to objects by decreasing the distance between their sensors (eyes) and the attended object.

In other related work [Orhan et al., 2020, Orhan and Lake, 2023], the authors trained self-supervised learning models from scratch on the SAYCam dataset [Sullivan et al., 2021], an egocentric video dataset captured by children between 6-32 months old. They showed that deep SSL models can achieve powerful visual representations and reasonable performance while solely trained on child-egocentric videos. [Orhan et al., 2020] used a temporal classification objective, which can be regarded as an implementation of the principle of temporal slowness [Wiskott and Sejnowski, 2002] by creating similar embeddings for the frames that were close in time and dissimilar embeddings for the frames that were far apart in time. To provide a more robust baseline, [Orhan and Lake, 2023] used a wider variety of SSL algorithms and architectures and found performance rates as high as 70% of a high-performing ImageNet-trained model. Furthermore, [Zhuang et al., 2021] showed that CNNs trained using contrastive learning on child egocentric videos can produce brain-like neural responses.

Although the previous work clearly shows that the structure in data sets matters and demonstrates the potential value of taking inspiration from human development, the data sets do not capture or

---

[1]https://cogdev.sitehost.iu.edu/homeview.html

exploit the *ordering of experiences* that is the hallmark of human developmental processes. In the prior cases, the datasets were created by aggregating training videos across large age spans (6 to 32 months), making it unclear what ages or aspects of the training data were responsible for the learning outcomes. In humans, the first year post-birth is a period of substantial developmental change in the cortical visual system, in visual sensitivities, and in object recognition [Jayaraman et al., 2015, Fausey et al., 2016]. Further, the earliest period – the first 3 to 4 months - constitutes a sensitive period in which disruptions in the expected visual input lead to lifelong disruptions in object recognition and other visual processes [Maurer, 2017].

Accordingly, we take an explicitly developmental approach [Smith et al., 2018] and show that doing so has a profound impact on self-supervised learning. Using a corpus of images from infants 2 to 12 months of age, we show that training that places the natural earliest visual input first optimizes learning. Training with data out of the natural developmental order disrupts learning. To demonstrate the generality of this ordering effect, we used self-supervised pre-training across a variety of algorithms and architectures; in all cases, a developmentally ordered visual diet led to the strongest learning outcomes.

## 2.2 Curriculum Learning in AI

In curriculum learning, models undergo training by initially being exposed to simpler examples and progressing to more difficult ones [Elman, 1993, Bengio et al., 2009, Wu et al., 2020]. The approach, common in education, arranges information in an orderly manner to enhance learning [Smith et al., 2018, Avrahami et al., 1997].

The curriculum contributes to learning by shaping the loss landscape. A good curriculum will result in a smoother learning trajectory. A substantial body of literature in AI focuses on designing ways to order the training samples based on some definition of sample difficulty to allow the models to converge faster with better downstream performance [Wu et al., 2020, Khan et al., 2023, Wang et al., 2019, Guo et al., 2018, Liu et al., 2017]. Unlike human and animal learning [Wilson et al., 2019], the extent of the impact of curricula on standard, offline, supervised learning is debated [Wu et al., 2020, Saglietti et al., 2022]. However, recent work shows a remarkable impact from curricula in online, self-supervised learning settings [Saglietti et al., 2022, Joshi and Mirzasoleiman, 2023].

In this work, we show the impact of the infant visual curriculum, even when using common, offline, self-supervised learning methods. This curriculum is not based on an a priori definition of sample difficulty, but rather assumes the natural ordering achieved through human development, an ordering that leads to the full prowess of human object recognition [Smith and Slone, 2017, Zaadnoordijk et al., 2022].

## 2.3 Computational neuroscience work on learning visual features from naturalistic stimuli

Early seminal work by [Olshausen and Field, 1996] demonstrated that sparse coding algorithms, when trained on natural images, produced receptive fields similar to those found in the primary visual cortex, suggesting a potential computational principle underlying visual learning in the brain. Similarly, work by [Bell and Sejnowski, 1997] showed how independent component analysis could be used to derive Gabor-like filters, again resembling the response properties of neurons in the visual cortex. More recent work [Karklin and Lewicki, 2009] has built on these principles, demonstrating how more complex visual features, like edge co-occurrence statistics, can also be learned from naturalistic stimuli. These studies collectively highlight the importance of naturalistic stimuli in shaping our understanding of the computational principles underlying visual learning in the brain.

In this study, for the first time, we explore the learning of visual features from the embodied data streams produced by infants at different stages of development, including infants younger than six months of age.

## 3 Datasets

### 3.1 Training Dataset

The datasets were created from a larger corpus of 500 hours of head camera videos collected by 101 infants from 2 to 24 months of age [Fausey et al., 2016]. From this corpus, we selected videos

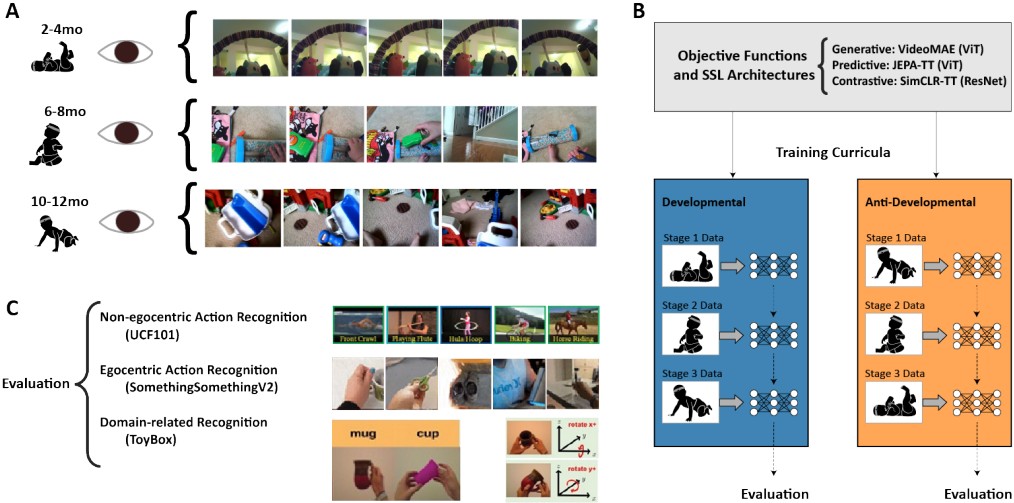

Figure 1: **Methods overview**. **A)** Sample image sequences from each infant age group in the training dataset. **B)** Training procedure: Each model is trained in three stages using a self-supervised objective function. **C)** Evaluation: After training, the learned representations are evaluated for their informativeness in three downstream tasks.

collected by infants at 3 age periods who differ in motor skills, visual preferences, and behaviors and, as a consequence, in their statistics of experience. The youngest infants were 2-4 months old (n = 10, total of 41 hours, mean age = 2.9 mo, SD = 0.75), a period characterized by limited postural control; the middle group of infants were 6-8 months (n = 10, total of 42 hours, mean age = 6.9 mo, SD = 0.62) and were mastering sitting and crawling, and the instrumental use of objects; and the oldest group of infants, 10-12 months (n = 9, a total of 40 hours, mean age = 11.1 mo, SD = 0.65), were beginning to walk, could sit stably and with increasing skill in manipulating objects. In the rest of the manuscript, we refer to these three age groups as $g0$, $g1$, and $g2$, respectively. See Supplemental Information, Table S3 for recording information per participant.

We used two additional training sets: (1) Egocentric images collected by adults in the home (18 to 71 years of age, 25 hours of recording) using the same camera, here referred to as $adult$, and (2) a random age infant dataset, referred to as $rand$, constructed by combining a random set of 13 hours of recordings from each infant in the three age groups $g0$, $g1$, and $g2$.

All video files were subsampled to RGB images at 30 frames per second. Pretraining on a given dataset started by selecting 1.45 million consecutive frames from the given age group (upon concatenating the frames from all subjects within the group). From the created sets of frames, we collected sample sequences of size 16 as inputs for the VideoMAE model (81,000 samples for the training set and 9,000 samples for the validation set). For JEPA-TT and SimCLR-TT, we used pairs of consecutive frames that are 10-30 seconds apart. The number of samples was kept fixed for all age groups and conditions. To allow the data-related variability to be reflected in the results, each dataset was split into 3 folds, and each pretraining run was performed using a different fold.

## 3.2 Benchmark Datasets

**UCF101** [Soomro et al., 2012] is a widely used dataset for video-based action recognition research. The dataset consists of 13,320 videos of realistic actions collected from YouTube. These videos are divided into 101 action categories, such as playing a flute, biking, and horseback riding. Each category contains between 100 to 200 video clips, which are approximately 7 seconds long on average. The wide variety of actions and the realistic setting make the UCF101 dataset a challenging benchmark for evaluating the performance of video understanding algorithms. This dataset is commonly used for evaluating the quality of spatiotemporal features in video models [Tran et al., 2015, Xie et al., 2017].

**Something-Something V2 (SSv2)** [Goyal et al., 2017] is an egocentric video dataset of short videos comprising humans performing pre-defined, basic actions with everyday objects. Being an egocentric dataset, SSv2 is more appropriate than UCF101 for evaluating our infant-egocentric models. In addition, SSv2 emphasizes the importance of dynamic features more than UCF101, which is known to be solvable by only attending to static features of individual frames. Here we evaluate our models on the 10-class subset of the dataset (as suggested in the dataset article).

**ToyBox** [Wang et al., 2018] is a small but developmentally relevant dataset of 4320 videos, 2-20 seconds each. In each video, an object undergoes a spatial transformation, such as rotation along the x-axis or translation along the z-axis (12 transformations total). There are 12 toy object categories (4 from animals, 4 from household, 4 from vehicles), each containing 30 distinct objects. The existence of both object categories and spatial transformations allows us to probe the representations learned by the models in terms of both shape-relevant and location/pose-relevant information.

## 4    Models and Objectives

Evaluating the impact of a developmental visual curriculum requires (1) a model with a self-supervised learning objective [Schiappa et al., 2022] that can scale up to a large corpus of real-world egocentric videos and (2) an architecture that directly learns regularities in videos across both space and time: a spatiotemporal learner. Generative and predictive tasks such as masked modeling (e.g. VideoMAE [Tong et al., 2022] or MaskFeat [Wei et al., 2022]) are particularly suitable for this purpose, although other methods based on cross-modal agreement or contrastive learning can be used as well.

### 4.1    Generative Vision Transformer (MAE)

Transformer-based masked autoencoders are effective and highly scalable methods of representation learning with minimal inductive biases. They are applicable to various sensory modalities, including both images and videos, and are therefore great candidates for a general spatiotemporal learner [Feichtenhofer et al., 2022]. Here we chose VideoMAE as a simple adaptation of the masked modeling method for videos. In this learning method, each sample video is broken down into a sequence of spatiotemporal tubes. During pretraining, a portion of the tokens are randomly chosen and masked, and the pretraining objective is to recover the masked tokens. This is achieved using a series of multi-head attention layers [Dosovitskiy et al., 2020] which need to learn the relationship between the tokens in order to successfully predict the masked tokens. From an intuitive perspective, each token can be thought of as a simple dynamic feature (e.g., akin to motion energy features) corresponding to a small area in the visual field; the entire video is the collection of many such features. The mask prediction objective promotes the network to learn the spatiotemporal organization of simple features transforming into complex moving features that occur in the dataset videos.

### 4.2    Predictive Vision Transformer (JEPA-TT)

I-JEPA [Assran et al., 2023] is a recently proposed image-based SSL algorithm in which the objective is to predict the embedding of one part of the input image (target) using the embedding of another part (context). Therefore unlike masked autoencoders, this objective function does not require pixel level reconstruction. Our JEPA Through Time (JEPA-TT) algorithm is an adaptation of I-JEPA, where the context and the target are selected from consecutive video frames rather than the same frame.

### 4.3    Contrastive CNN (SimCLR-TT)

SimCLR [Chen et al., 2020] is a well-known image-based SSL algorithm. It works by bringing the embeddings of two augmented versions of an image (positive pair) closer in the embedding space while pushing them away from embeddings of randomly selected images from the dataset. Similar to Aubret et al. [2022], our SimCLR-TT algorithm treats consecutive frames as augmentations of each other. To test additional network architectures, we use a convolutional backbone, ResNet-18, for SimCLR-TT.

# 5 Evaluation Criteria

## 5.1 Downstream Performance

After pretraining each model, the encoder part of the model was evaluated in several downstream tasks using both linear classification and k nearest-neighbor (kNN) classification. We first collected the encoder responses to all stimuli in the downstream video dataset as feature vectors. For linear classification, we evaluated the usefulness of the features by training and validating a linear SGD classifier. For kNN classification (results shown in the Supplemental Information), for each sample embedding in the validation set, we first found the k nearest samples in the training set. We then evaluated whether the class of any of the k nearest neighbors matched the true class of the validation sample. All results in the pretraining and downstream evaluation were generated with 3 seeds per curriculum condition.

## 5.2 Gradient Norm

In order to provide a glimpse into the learning process that gives rise to the downstream performance results, we report the Stochastic Gradient Descent (SGD) gradient norms across the training stages. We compute gradient norm of a given model layer as $||\frac{1}{batchsize} \sum \frac{Loss}{\partial \Theta}||_2$ where $\Theta$ contains the learnable parameters of the layer of interest and the sum goes over all the samples in the batch. Learning in each parameter is driven by the magnitude and direction of the noisy, empirical gradient estimate after being averaged across batch samples. This quantity has been used for studying the learning dynamics of SGD [Hochreiter, 1998, Shwartz-Ziv and Tishby, 2017]. Gradient norms can vary as a function of the architecture, optimizer, and objective function and hence, should be interpreted carefully. Here we evaluated the effect of the order of the visual diet on the gradient norm by matching all other system components. We also reproduced the results in various architectures and objective functions.

# 6 Results

## 6.1 *Stage 1*: Visual inputs from the youngest infants lead to stronger learning signals and better downstream classification

To characterize the impact of training with infant data from different stages of development, we first tracked the gradient norm of the VideoMAE model trained separately with the data from the youngest age group ($g0$), the oldest age group ($g2$), the random group (see Section 3.1), and the adult group. Training with the data from the youngest age group provided the largest gradient norm in the model (shown for the last encoder layer in Figure 2 panel A)

Next, we evaluated the performance of the VideoMAE pretrained in *stage 1* on a downstream classification task. As described in Section 3.2, we used the Something-Something v2 (10 classes), UCF-101, and Toybox datasets. For Toybox, we compared pretrained models on spatial transformation classification. In all cases, the model pretrained on the data from the youngest age group performed substantially better than other models (Figure 2 panels B-D). These results confirm that models that learn from larger learning signals (gradient norms) during the pretraining stage extract more relevant features from video data. This was true across domains since object classification, spatial transformation classification, and video classification are substantially different tasks.

## 6.2 *Stage 1-3*: Developmental curriculum matters

We examined whether the developmental curriculum contributes to larger gradient norms and better downstream classification. After pretraining VideoMAE on *stage 1* using different age groups, we continued the pretraining process with two more stages. In one case, we followed the developmental curriculum by training on data from 2-4 months old infants ($g0$) in *stage 1*, data from 6-8 month old infants ($g1$) in *stage 2*, and then data from the oldest, 10-12 month, age group ($g2$) in *stage 3*. Each stage had 5,000 iterations. For comparison, we also pretrained models in an anti-developmental curriculum order by training on $g2$ in *stage 1*, $g1$ in *stage 2*, and $g0$ in *stage 3*. Similar to pretraining on just *stage 1*, we also pretrained the network on the random group, but this time across all three

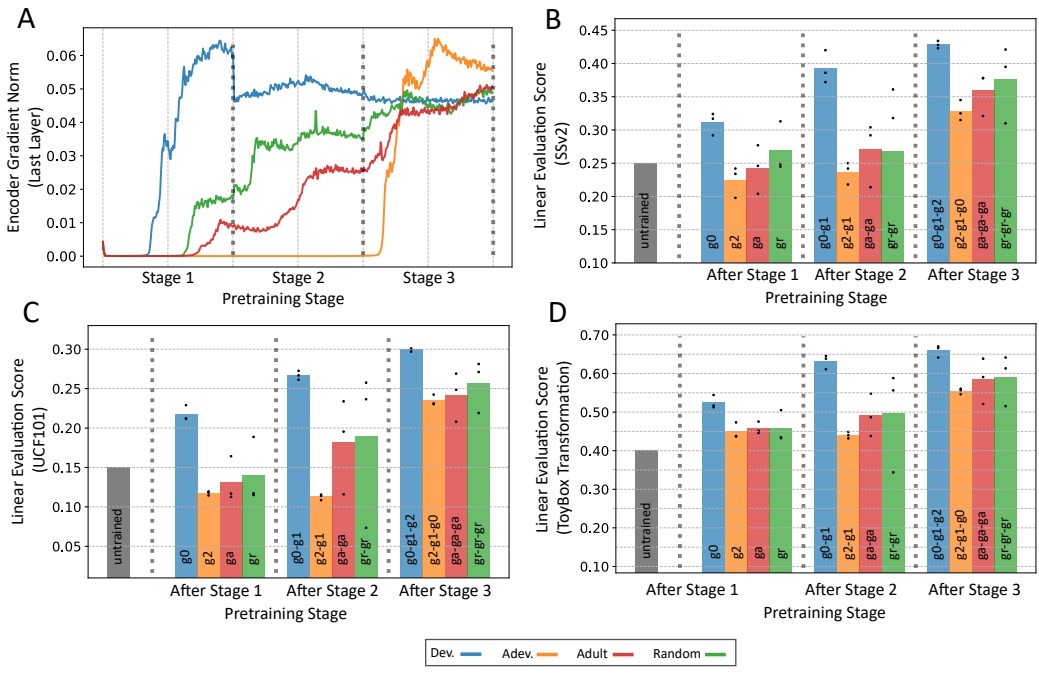

Figure 2: **The effect of learning curriculum.** A) The norm of the gradient at the last layer of the VideoMAE encoder across training iterations. To increase visibility, the lines have been smoothened by applying a moving average with a window length of 100. The average line across 3 seeds is plotted. B-D) The linear evaluation score of the pretrained VideoMAE models on downstream video classification tasks of SomethingSomethingV2-10class, UCF101, and ToyBox object transformation. Each black dot indicates the result from one seed. The performance of an untrained VideoMAE model is shown in gray.

stages. Finally, we pretrained on adult data in *stage 1* followed by $g1$ and $g2$ in *stage 2* and *stage 3*, respectively.

As shown in Figure 2, training in developmental order (*blue bars*) led to the highest performance in the subsequent classification tasks. Anti-developmental curriculum (*orange bars*) led to the lowest downstream performance. This is an important result since data from $g0$ were used in both the developmental and the anti-developmental curricula. However, because $g0$ was used at the last stage of pretraining for the anti-developmental curriculum, the performance gain was less than when $g0$ was used in *stage 1*. This underscores the importance of training data arriving in the correct developmental order.

### 6.3 Developmental advantage comes from the temporal slowness and simplicity of the views acquired by young infants

In this subsection, we ask why training with young infant data leads to better results. Previous work shows that the visual experience of young infants is substantially slower than that of older infants [Sheybani et al., 2023]. We hypothesize that the advantage of the developmental curriculum is due to this temporal slowness. A considerable literature also shows that young infants' immature visual system has low acuity [Lewis and Maurer, 2005] and as a result, in laboratory studies, young infants prefer to look at arrays with high contrast and a few edges [Kiorpes, 2016], that is scenes that are spatially simple. To test this hypothesis, we created *simplicity-matched* subsets of the data from the different age groups: To match temporal complexity, we created samples that consist of only one frame. To match spatial complexity, we selected a subset of frames from each age group where around 6% of frame pixels were part of an edge, as computed using the Canny edge detector. We then repeated our computational experiment in the last section, using the newly created version of the developmental and anti-developmental curricula (Figure 3).

As predicted, reducing the spatial complexity (Matched Spatial and Matched Spatiotemporal) as well as training on single frames (Matched Temporal and Matched Spatiotemporal) increased the gradient norm in both curricula (Figure 3 panel A). This increase in the gradient norms is echoed in the downstream performance of the models within the same stage and curriculum. Comparing the gap between the developmental and anti-developmental curricula, we find both metrics of gradient norm and downstream performance largely reduce as we match the spatial complexity, temporal complexity, or both. The aforementioned observations confirm the advantage of early temporal and spatial simplicity in the self-supervised learning models. We finally note that the models trained on single frames had diminished performance on SSv2, demonstrating the value of the motion information across multiple frames (Figure 3 panel B).

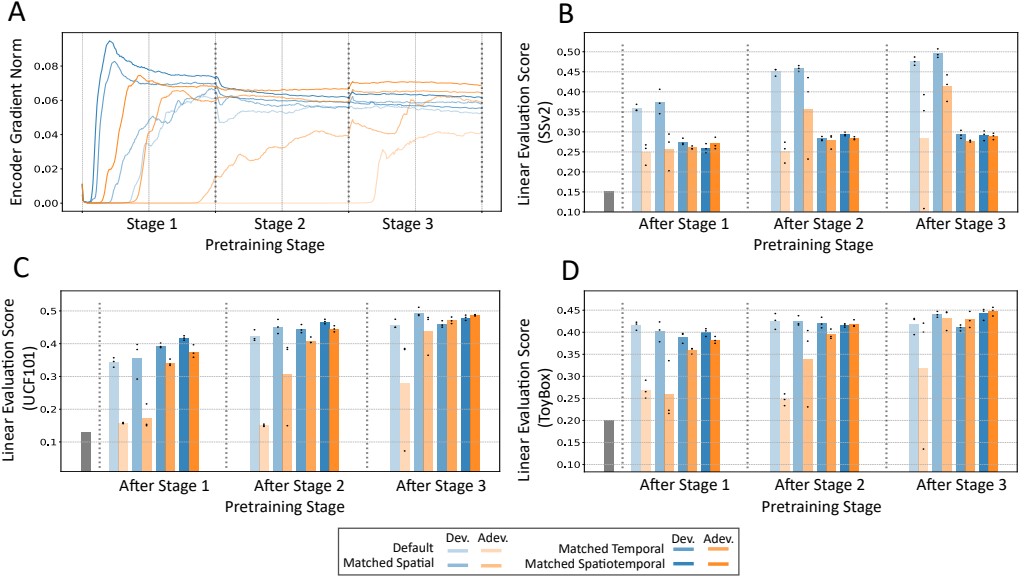

Figure 3: **The effect of the learning curriculum before and after controlling for spatial and temporal complexity.** The curricula are indicated by color hue and the conditions are indicated by lightness. A) The norm of the gradient at the last layer of the VideoMAE encoder across training iterations. To increase visibility, the lines have been smoothened by applying a moving average with a window length of 100. The average line across 3 seeds is plotted. B-D) the linear evaluation score of pretrained VideoMAE models on downstream video classification tasks of SomethingSomethingV2-10class, UCF101, and ToyBox object classification. The performance of an untrained VideoMAE model is shown in gray.

## 6.4 Curriculum effects hold for other learning objectives

In this subsection, we asked whether the curriculum effect is limited to the generative learning objective or the vision transformer architecture. Several computer vision algorithms have been considered by the computational neuroscience community as potential algorithmic hypotheses for human perceptual learning, with no consensus as of yet. We previously focused on VideoMAE due to its direct interaction with the spatial and temporal complexity of the input, without potential difficulties of predictive algorithms, such as representation collapse. Here we reproduce our main result using two more self-supervised learning algorithms: JEPA-TT with a vision transformer backbone and SimCLR-TT with a convolutional backbone (see the Methods section for implementation details). Both of these algorithms take single images as inputs and, therefore, only indirectly interact with the temporal continuity of the input. However, they can still benefit from the spatial simplicity of young infant views and may also indirectly benefit from temporal slowness because of the similarity of the image pairs that are close in time.

Figure 4 shows that the advantage of the developmental curriculum holds for both JEPA-TT and SimCLR-TT. We observe that the effect size is smaller than that of VideoMAE, which may be due to the indirect interaction of these algorithms with the spatiotemporal complexity of their data stream.

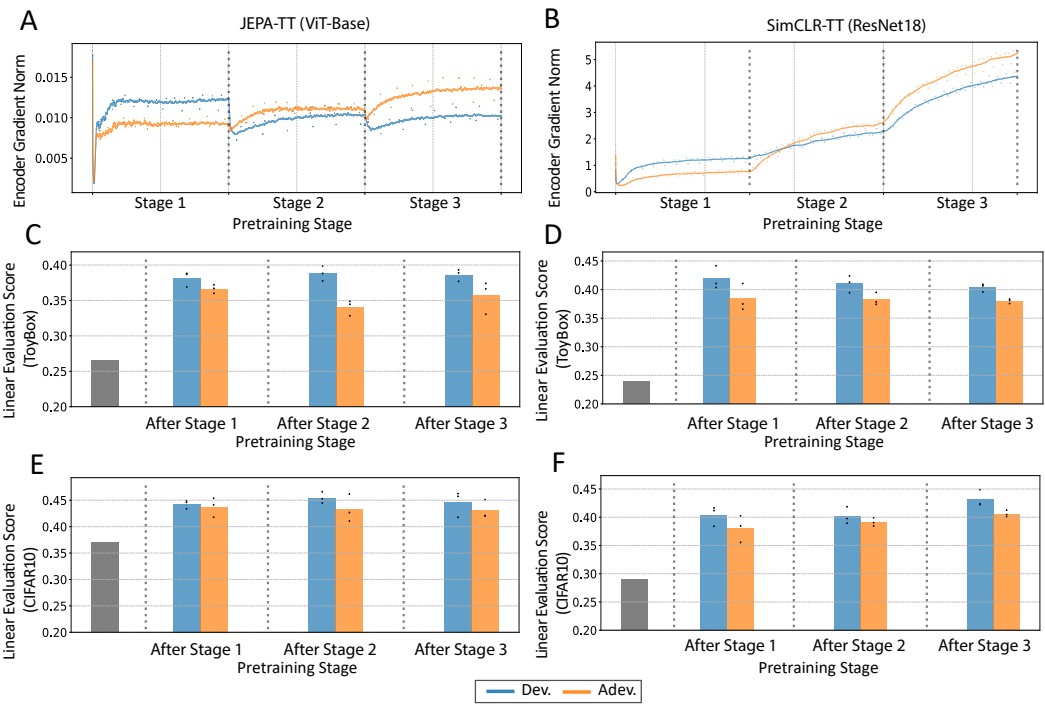

Figure 4: **The effect of the learning curriculum on two CLTT learning systems (JEPA-TT w/ ViT-Base backbone), SimCLR-TT w/ ResNet18 backbone).** The curricula are indicated by color hue, with blue for Developmental and orange for Anti-Developmental. A,B) The gradient norm of the last layer of the encoder. C-F) The linear evaluation score of the pretrained models on downstream image classification tasks. The performance of an untrained model is shown in gray.

## 6.5 Ablation Analysis

To determine the robustness of the results, we conducted several ablations studies. We showed that the qualitative pattern of results did not change when we used a different optimizer (Figure S3), or a different number of pretraining iterations (Figure S5). Similar to [Wu et al., 2020, Saglietti et al., 2022], we observed that the importance of curriculum order decreases as the number of training iterations and epochs increases.

## 7 Discussion

Our findings make three contributions. First, they provide computationally explicit evidence that human infants generate high quality data for visual learning across the first year of development that changes in a developmentally ordered way in its underlying properties. Critically, we show that the order of the data matters. Deep neural networks trained on developmentally ordered datasets outperformed those trained on "developmentally incorrect" datasets. Second, the findings indicate that the properties of the initial training set are particularly critical. This result accords with studies showing that the first few months after birth constitute a critical period for human visual development, with disruptions in the expected input leading to lifelong consequences for vision and visual cognition [Maurer, 2017]. The relative importance of the earliest training set in these computational models offers new insights into human visual development. Critical periods in biological development are typically thought to emerge from changes in the brain rather than from changes in experience. The current findings suggest that the properties of earliest experiences in and of themselves constrain what can be learned from later experiences in infants and machine learning. Third, the findings indicate a potential principle for what constitutes an optimal starting point for visual learning: Simplicity in spatial structure of the images and in the slowness of change in those images.

Recently Joshi and Mirzasoleiman [2023] showed that data samples that contribute the most to self-supervised learning are those that have the most similar augmentations to the other samples. Such samples are shown to ensure a high alignment between the augmented views of the same class, and preserve the centers of class representations. This suggests a possible computational mechanism for how slowness and spatial simplicity benefit SSL: by providing more similar augmentations of the same object (and class). This may be particularly beneficial at the start of training by stabilizing and distinguishing the representation of objects in the high-dimensional space. The constraints of physical space and time on infant experiences mean that real-world experience is extremely skewed to many experiences of just a few individual instances. Infants, for example, do not see many instances from a given object class early on, e.g. not many different cups, but rather they see one cup frequently [Clerkin et al., 2017]. Smith et al. [2018] discussed several verbal hypotheses of how this property of the infant experience can facilitate or affect visual learning. A noteworthy account in accordance with [Joshi and Mirzasoleiman, 2023] is that early learning in both infants and SSL models can be facilitated by learning to see fewer objects but in many different viewing conditions. We note that the notion of class centroids in [Joshi and Mirzasoleiman, 2023] closely resembles the concept of prototype in the prototype theory of human perceptual categorization [Rosch and Mervis, 1975]. This is another indication that self-supervised deep ANNs can appropriately function as a continuation of the classic connectionist models in the human perceptual learning literature [Way, 1997, Doerig et al., 2023].

Traditional theories of visual development also do not provide a quantitative account of how newborn visual systems learn from raw visual inputs. As such, these theories do not explain how or why newborn visual systems develop the way that they do. By training deep neural networks "through the eyes" of developing humans, we can tackle these problems and begin building neurally mechanistic, image-computable models of human newborn visual systems.

### 7.1   Limitations and Future Directions

Our study focused on showing the effect of the learning curriculum on the learning outcomes, which are likely related to the learning dynamics of SGD networks, particularly those learning from self-supervised errors. We did not aim for the models trained here to perform competitively on any benchmark of neural or behavioral match to humans. Nevertheless, a promising future direction for this work is to aim for *milestone models of development* – models that develop the same visual skills as infants at a particular stage given the same visual experience. This endeavor involves developing benchmarks of infant behavioral and neural responses in machine learning tasks, similar to [Schrimpf et al., 2021, Geirhos et al., 2021] as well as making engineering progress in continual and lifelong learning [Zhuang et al., 2022].

Upon the remarkable success of the neuroconnectionist research program [Doerig et al., 2023], there have been calls for bridging visual developmental neuroscience and deep learning [Mur, 2023, Zaadnoordijk et al., 2022]. We argue that work along the aforementioned lines can take a remarkable step towards a computational understanding of the developmental trajectory of the human visual system while also providing a road map for training computer vision systems more efficiently.

## Acknowledgement

The authors thank Kathryn Bonnen, T. Rowan Candy, David Crandall and the other members of the egocentric vision group at Indiana University for the inspiring discussions that helped develop the ideas used in this project. The authors acknowledge the Indiana University Pervasive Technology Institute [Stewart et al., 2017] for providing supercomputing and storage resources that have contributed to the research results reported within this paper. This research was supported in part by Lilly Endowment, Inc., through its support for the Indiana University Pervasive Technology Institute. The collection of the dataset used in this project was supported by NIH grant 1R01EY032897 to Linda Smith and T. Rowan Candy.

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
