# A    Supplemental Information

## A.1    Pretraining Implementation Notes

For all models, the preprocessing of the input frames involved resizing the images with the short side as 224, followed by cropping the central patch of 224x224. Each color channel was then normalized to roughly follow a distribution with a zero mean and unit standard deviation.

The VideoMAE models were built using the off-the-shelf implementation from HuggingFace [Wolf et al., 2020] with the ViT Base backbone and a 90% mask ratio. To save computational resources, the number of ViT layers were reduced from 12 to 6, but all other hyperparameters were kept as default.

The JEPA-TT models were implemented by adapting I-JEPA [Assran et al., 2023] to use time as a source of augmentation. Therefore, the target patches and the context patch came from two separate frames with a 10-30s lag in between them. This choice of the time interval is informed by the temporal redundancy in the videos. Vit-Base was again used as the backbone architecture.

Similar to JEPA-TT, the SimCLR-TT models used time as a source of augmentation with the positive sample having a 10-30 second lag from the anchor sample. We found SimCLR-TT to be prone to overfitting on our infant dataset therefore we applied additional data augmentations and used fewer training iterations (1500 iterations per stage). The backbone for the SimCLR-TT models was ResNet-18.

All models were trained using the vanilla SGD optimizer (except for the ablation experiment on the optimizer) on 4 NVIDIA A100 GPUs. Each experimental run finished within 24 hours or less. The code is available at `https://github.com/ssheybani/baby-vision-curriculum`.

## A.2    Probing the Input Layer

Aiming to understand the properties of the dataset that lead to the advantage of the developmental curriculum, we performed PCA on the spatiotemporal patches from the videos in different age groups. This is similar to the procedure in Benjamin et al. [2022], but instead of forming the data matrix using the flattened version of a single frame (spatial patch), we used the flattened version of 2 consecutive frame patches, the same size as the spatiotemporal patches in our VideoMAE models. The resulting PCA bases provide a basis set for reconstructing all spatiotemporal patches in the dataset. In the case of single frames, the bases are static edge detectors at various angles and frequencies [Benjamin et al., 2022]. In our case, the resulting bases are detectors of *moving* edges since the inputs are patches from consecutive frames.

Figure S1A shows PCA bases of the spatiotemporal patches compared between $g0$ and $g2$. Compared to the bases from $g0$, those in $g2$ are distorted or do not appear as sharp. This is confirmed by analyzing the cumulative explained variance ratio (CEVR) (Figure S1B). The bases learned from $g0$ reconstruct its input with fewer components, indicating a sparser, more efficient coding scheme. The first 12 components of $g0$ explained 95% of the variance (5% error). It takes many more – 33 components– from $g2$ to reach 5% error rate.

We also repeated the analysis with a control condition by computing the PCA from individual frames from each group. The resulting PCA bases were indistinguishable in terms of their qualitative appearance and CEVR (Figure S2).

## A.3    Evaluation results using nearest neighbor search

There are a variety of methods to evaluate how useful the representations of a model are. By training a linear classifier on the features, one can assess how separable the benchmark samples are according to various target definitions, e.g. object category, object motion, etc.

Another common method for representation evaluation is retrieval performance or nearest neighbor evaluation. This method evaluates to what extent the target label of a sample can be inferred correctly using the label of closest samples in the embedding space. Tables S1, S2 show the comparison of various curricula using this method on the SSv2 and UCF101 benchmarks. The results remain consistent with those from linear evaluation.

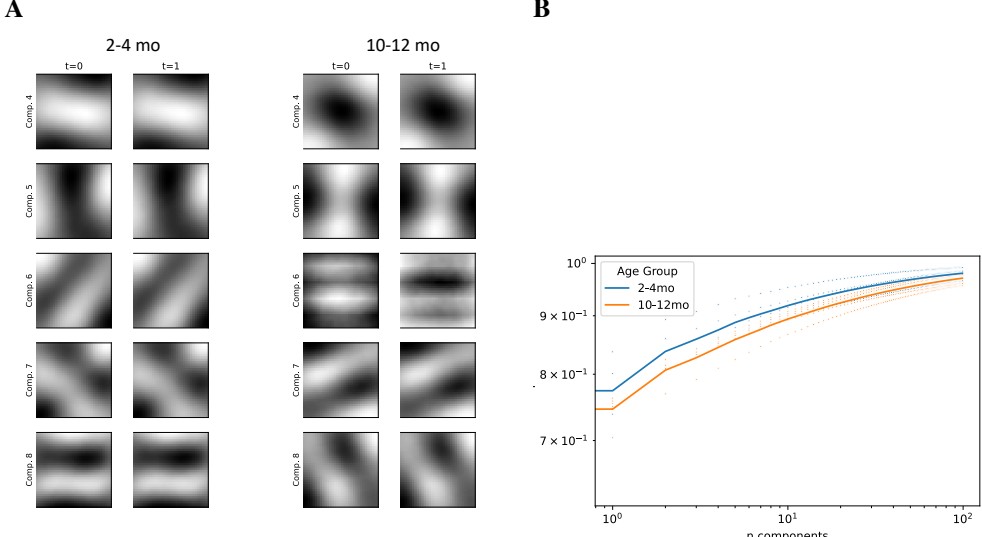

Figure S1: **A.** Samples from the dynamic PCA basis set of the data streams of the youngest and oldest infant groups. **B.** Cumulative explained variance ratio as a function of the number of PCA components.

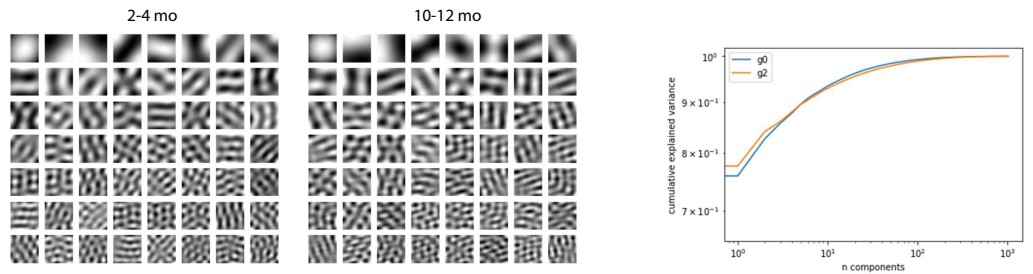

Figure S2: Control: Samples from the Static PCA Basis set of the Data Streams

## A.4 Details of the ablation analysis

We performed ablation studies on three key hyperparameters of the learning method: choice of optimizer (Figure S3), frame rate (Figure S4), and the number of training iterations at each stage of curriculum pretraining (Figure S5).

Our results, i.e., the advantage of the developmentally-ordered curriculum over the opposite order, remain qualitatively the same in all conditions, both in terms of convergence rate and downstream performance on Toybox object categorization.

The choice of the optimizer made the largest impact on the convergence rate (Figure S3A). Note that the pretraining loss is computed over the images from the same stage (i.e., at the last stage, *g2g1g0* loss is on *g0*, but *g0g1g2* loss is on *g2*). Therefore even though the final pretraining loss of *g2g1g0* in the *adamw* condition is lower than that of *g0g1g2*, this cannot be interpreted as an advantage for *g2g1g0*. Aligned with this, we see comparable downstream performance for the two curricula in the *adamw* condition. Nevertheless, we conjecture that, in general, the difference in the convergence rate between curricula is due to the different extent of ruggedness in the loss landscapes shaped by the data streams of the different age groups. ADAM and ADAMW are engineered to navigate such landscapes better than vanilla SGD. However, the extra bells and whistles in the ADAM and ADAMW may or may not make them closer to human visual learning compared to vanilla SGD.

We also examined the effects of having different numbers of training iterations per pretraining stage. We saw that by allowing many iterations on *g0* at the final stage, the anti-developmental

Table S1: Performance comparison of the curricula on SSv2 video retrieval (nearest neighbor search)

| Stage | Curriculum | SSv2 Top1 | SSv2 Top5 | SSv2 Top10 |
|---|---|---|---|---|
| 0 | Untrained | 10.10 | 42.00 | 64.40 |
| 1 | Dev. | **14.90** | **48.20** | **68.47** |
|  | Adev. | 10.07 | 42.67 | 64.27 |
|  | Random | 11.67 | 44.50 | 65.50 |
|  | Adult | 10.47 | 42.90 | 64.33 |
| 2 | Dev. | **21.33** | **58.10** | **75.10** |
|  | Adev. | 10.70 | 42.10 | 63.90 |
|  | Random | 16.53 | 49.23 | 70.13 |
|  | Adult | 14.30 | 46.03 | 68.67 |
| 3 | Dev. | **21.80** | **59.60** | **77.17** |
|  | Adev. | 17.70 | 51.80 | 70.57 |
|  | Random | 20.70 | 55.97 | 73.97 |
|  | Adult | 19.10 | 54.50 | 74.00 |

Table S2: Performance comparison of the curricula on UCF101 video retrieval (nearest neighbor search)

| Stage | Curriculum | UCF101 Top1 | UCF101 Top5 | UCF101 Top10 |
|---|---|---|---|---|
| 0 | Untrained | 10.67 | 20.02 | 26.86 |
| 1 | Dev. | **19.29** | **31.35** | **38.91** |
|  | Adev. | 10.70 | 20.18 | 27.04 |
|  | Random | 12.53 | 22.39 | 29.63 |
|  | Adult | 12.31 | 21.86 | 29.11 |
| 2 | Dev. | **22.27** | **36.36** | **45.00** |
|  | Adev. | 10.64 | 20.43 | 27.57 |
|  | Random | 17.33 | 28.68 | 36.86 |
|  | Adult | 15.75 | 26.67 | 33.96 |
| 3 | Dev. | **24.19** | **39.17** | **48.53** |
|  | Adev. | 20.83 | 33.11 | 40.83 |
|  | Random | 22.81 | 36.06 | 44.37 |
|  | Adult | 21.71 | 34.30 | 42.67 |

curriculum approaches the downstream performance of the developmentally-order curriculum. However, as in S4 the amount of reduction in loss is larger when the model begins the training with $g0$.

**A**

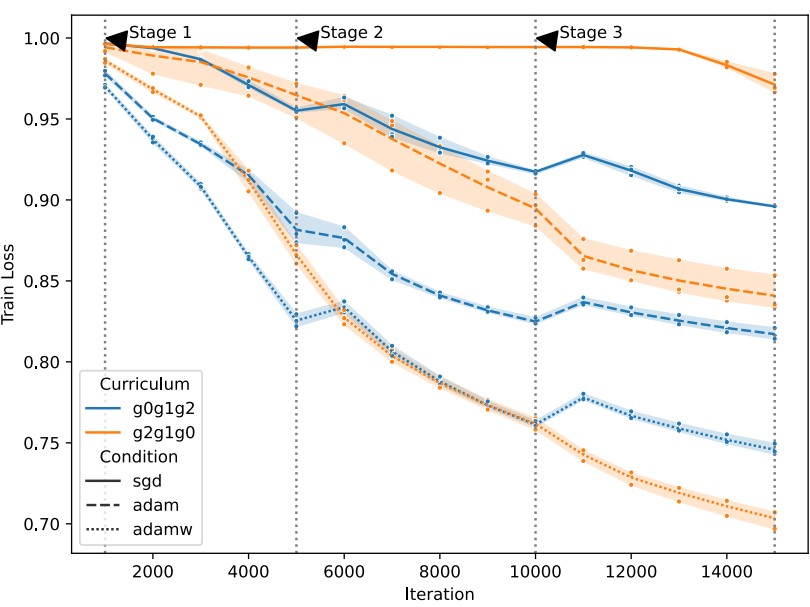

**B**

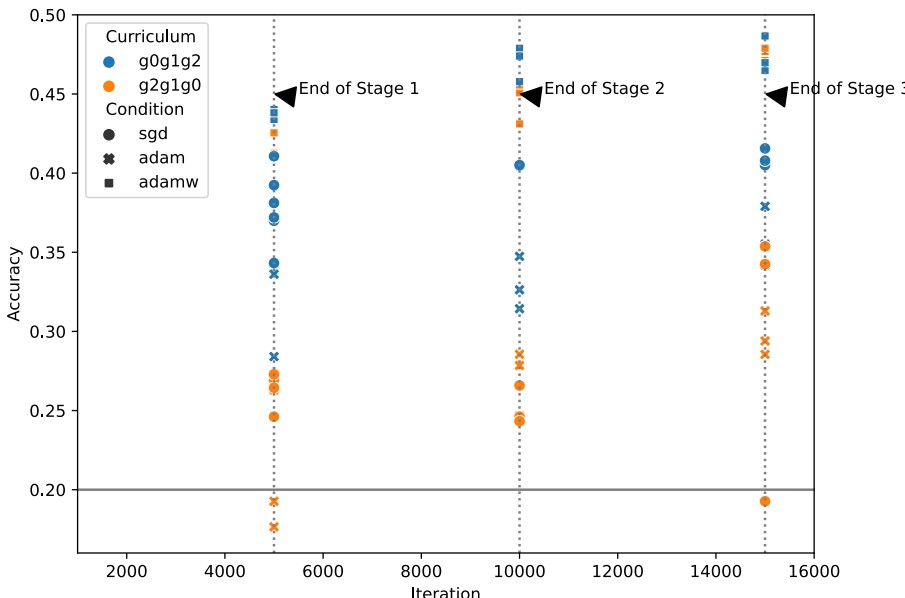

Figure S3: Pretraining with different optimizers **A.** Pretraining loss **B.** Toybox categorization accuracy

**A**

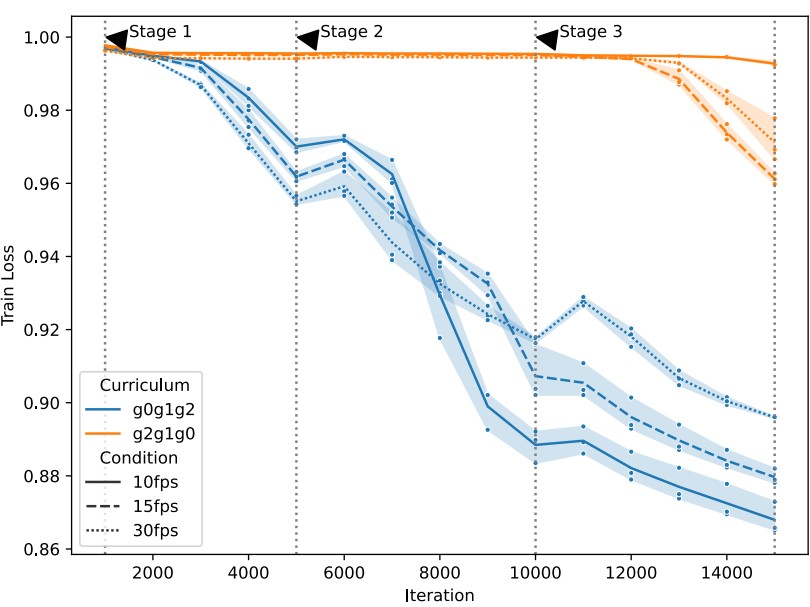

**B**

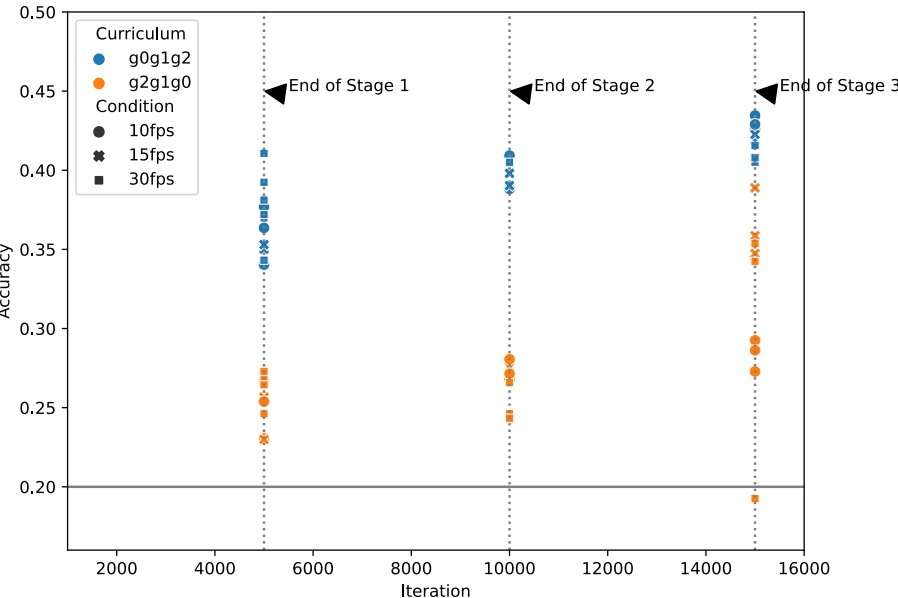

Figure S4: Pretraining with different frame rates **A.** Pretraining loss **B.** Toybox categorization accuracy

**A**

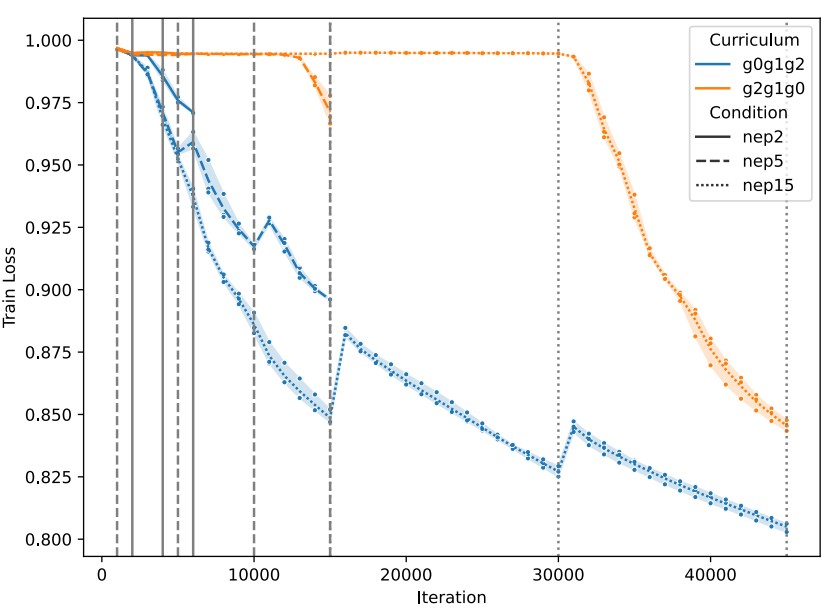

**B**

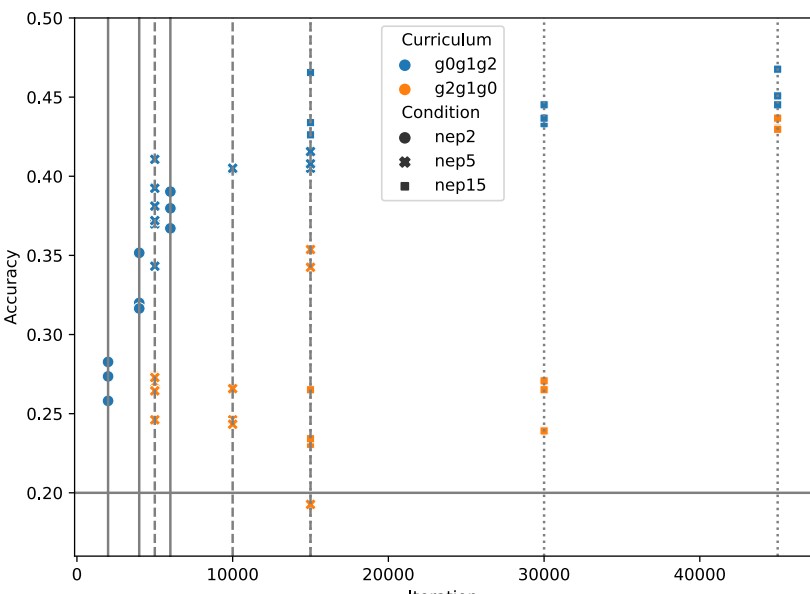

Figure S5: Pretraining with different numbers of epochs per stage **A.** Pretraining loss **B.** Toybox categorization accuracy

| Participant ID | Age (weeks) | Recording Duration (h) |
|---|---|---|
| MS | 8 | 4.45 |
| SS | 9 | 3.77 |
| BF | 10 | 3.22 |
| EA | 11 | 5.52 |
| TT | 12 | 5.12 |
| LS | 13 | 5.21 |
| SN | 14 | 6.32 |
| JM | 15 | 2.45 |
| TF | 16 | 5.44 |
| EW | 17 | 10.38 |
| | | |
| AR | 26 | 5.39 |
| SS | 27 | 5.47 |
| CK | 28 | 4.01 |
| MR | 28 | 4.62 |
| TT | 29 | 5.36 |
| FD | 30 | 4.51 |
| HW | 31 | 6.06 |
| SR | 32 | 4.97 |
| SE | 33 | 6.28 |
| JC | 34 | 4.04 |
| | | |
| MP | 43 | 2.2 |
| ET | 44 | 5.59 |
| TE | 46 | 7.05 |
| MS | 47 | 1.27 |
| KG | 48 | 4.53 |
| JC | 49 | 5.74 |
| AB | 50 | 5.85 |
| AK | 50 | 5.36 |
| DW | 51 | 4.14 |
| | | |
| BR | Adult | 2.62 |
| CW | Adult | 5.76 |
| EA | Adult | 1.06 |
| ED | Adult | 1.31 |
| JB | Adult | 1.41 |
| KI | Adult | 3.35 |
| LS | Adult | 4.46 |
| SB | Adult | 3.51 |
| TR | Adult | 3.64 |

Table S3: Participant info.

| Curriculum | 5000 | 10000 | 15000 |
|---|---|---|---|
| Dev. (g0-g1-g2) | 4.15 | 4.18 | 3.17 |
| Anti-Dev. (g2-g1-g0) | 0.21 | 0.019 | 2.32 |
| Random (rand-rand-rand) | 0.62 | 2.82 | 3.42 |
| Adult (adult-g1-g2) | 0.22 | 1.59 | 2.86 |

Table S4: Amount of reduction in the numerical values of self-supervised loss (100x) as a result of each stage of pretraining.