# OpenReview forum: "Curriculum Learning With Infant Egocentric Videos"
_NeurIPS.cc/2023/Conference — NeurIPS 2023 spotlight_

### Official Review · Reviewer_Bo4Q · 2023-07-02

**Soundness:** 3 good
**Presentation:** 3 good
**Contribution:** 3 good
**Rating:** 7
**Confidence:** 3

**Summary:**

This paper studies the development of visual intelligence by providing a curriculum for self-supervised video representation learning. The curriculum is aligned with the age order of the infants whose egocentric video data are used as training samples. Experiment results demonstrate that training with data from the youngest infants, who have fewer motor abilities, leads to a faster decrease in both training and validation loss. Additionally, the proposed curriculum improves downstream task performances when evaluated using linear probing.

**Strengths:**

- The use of curriculum learning guided video self-supervised learning to study infant visual experience is a novel and interesting idea.
- The natural alignment of the curriculum with the age order of infants, based on the egocentric videos recorded, is a commendable aspect of this work.
- The paper provides a detailed discussion of the contributions and limitations of the research.

**Weaknesses:**

- Regarding the “slowness” of the video, there lacks a quantitative measurement of it, as well as corresponding experiments regarding the curriculum that is aligned with the slowness of the video rather than the age order.

-  The iteration number 5000 might be not sufficient for models trained on videos with more dynamics to converge. While ablation studies on iteration numbers are presented in the supplementary material, it would be valuable to continue to increase the iteration number, and include a comparison with Random and Adult baselines for a more comprehensive analysis.

- It is recommended to report the video retrieval performance after pre-training with different curricula to provide a more comprehensive evaluation.

- In addition to masked image modeling, exploring contrastive learning-based self-supervised learning methods would enhance the scope of the study.

- Some minor issues include missing section reference number in line 212.

**Questions:**

- Is the PCA applied to both the training and validation video data in a specific age group?
- Figure S6 for CEVR seems to be missing in the supplementary material.

**Limitations:**

The authors have discussed the limitations of the evaluation method. It would be beneficial for the authors to discuss further the limitations related to the potential biases of the data collection process, or limitations in the egocentric video data recorded from infants.

---

> ### Author Rebuttal · Authors · 2023-08-09
>
> We thank the Reviewer for detailed comments and for highlighting the values of this line of work. Below we addressed each weakness and question mentioned by the Reviewer.
>
> **Reviewer** pointed out the lack of a quantitative measurement for “slowness” of the video, as well as corresponding experiments regarding the curriculum that is aligned with the slowness of the video rather than the age order.
>
> **Our response:** To better understand the role of temporal slowness in our results, we performed new analyses in which we controlled for spatial and temporal complexity (see general response for details). As shown in Fig. R1 (rebuttal PDF), temporal complexity was particularly important for driving the difference between developmental and anti-developmental curricula. Furthermore, we have conducted a detailed analysis of temporal slowness by estimating the autocorrelation function of the videos using the mean pairwise cosine similarity between the pixel representation of the frames at a series of lags. We then used the time constant of the autocorrelation function as a measure of the slowness of the video. We found significant differences between the median time constant across all three age groups, with the youngest age group (g0) being the slowest, followed by the middle age group (g1). The results were further confirmed through a spectral analysis of the embeddings, where the youngest age group had more low-frequency components than the other two age groups. We will add these results to the supplemental of the manuscript.
>
> **Reviewer** suggested to continue to increase the iteration number, and include a comparison with Random and Adult baselines for a more comprehensive analysis.
>
> **Our response:** Fig. S3 in the manuscript shows the pretraining and downstream classification results for up to 15000 interactions per stage. We did not observe qualitative changes in training between 2000, 5000 and 15000 iterations. Following the suggestion from the Reviewer, we are currently conducting experiments for 30000 iterations as well as for Random and Adult baselines. Due to computational complexity, these experiments will take several days to complete. We also note that the choice of 5000 iterations for each of the three stages of training was inspired by the Neyshabur et al. study, which reported the effect of curriculum going away if the total number of iterations reaches around 20k.
>
> **Reviewer** recommended to report the video retrieval performance after pre-training with different curricula.
>
> **Our response:** The reconstruction loss provided in Fig. 2 in the manuscript reflects the video retrieval performance. We apologize if we misunderstood the Reviewer’s suggestion.
>
> **Reviewer** suggested to explore contrastive learning-based self-supervised learning methods to enhance the scope of the study.
>
> **Our response:** We conducted additional evaluations using JEPA-TT and SimCLR-TT (see general response for details and Fig. R2 in the rebuttal PDF). Unlike VideoMAE, which is trained on reconstruction, JEPA-TT is trained on prediction using contrastive self-supervised learning through time. The results from JEPA-TT were qualitatively similar to the results from VideoMAE. SimCLR-TT is another approach based on contrastive learning (Aubret et al. 2022), which uses CNN instead of transformer architecture. We found again that the results remained qualitatively similar, supporting our general claim about the importance of the order of training data.
>
> **Reviewer** asked: Is the PCA applied to both the training and validation video data in a specific age group?
>
> **Our response:** Yes, that is correct.
>
> We thank the Reviewer for pointing out the missing section reference and that reference to Fig. S6 should point to Fig. S4.

---

> > ### Comment · Reviewer_Bo4Q · 2023-08-14
> >
> > I appreciate the authors' efforts to respond to all concerns raised by the reviewers. My questions are mostly addressed. The performance of video retrieval has been used to measure the quality of the learned representations in many video SSL methods, and the authors can refer to [1*, 2*] for more details.
> >
> > [1*] RSPNet: Relative Speed Perception for Unsupervised Video Representation Learning, AAAI 2021
> >
> > [2*] TransRank: Self-supervised Video Representation Learning via Ranking-based Transformation Recognition, CVPR 2022

---

> > > ### Author Response · Authors · 2023-08-17
> > >
> > > Thank you again for your feedback. We greatly appreciate your comments.
> > >
> > > **Training on 30k interactions per stage: Results remained consistent.**
> > > We are happy to report the results of training on a larger number of interactions (as we promised in the rebuttal). In the manuscript, we conducted experiments on 2.5k, 5k and 15k iterations per each of the three stages. Now we increased the number of interactions to 30k per stage (a total of 90k for three stages) and found that the results (provided in the table below) remained consistent.
> > >
> > > | Stage                  | Curriculum | SSv2 Accuracy | UCF101 Accuracy |
> > > |------------------------|------------|---------------|-----------------|
> > > | 1 (After Iteration 30k)| Dev.      | **0.458**         | **0.504**           |
> > > |                        | Adev.     | 0.293         | 0.208           |
> > > |                        | Random    | 0.413         | 0.387           |
> > > |                        | Adult     | 0.417         | 0.403           |
> > > | 2 (After Iteration 60k)| Dev.      | **0.523**         | **0.537**           |
> > > |                        | Adev.     | 0.487         | 0.471           |
> > > |                        | Random    | 0.499         | 0.510           |
> > > |                        | Adult     | 0.500         | 0.525           |
> > > | 3 (After Iteration 90k)| Dev.      | **0.553**         | **0.564**           |
> > > |                        | Adev.     | 0.497         | 0.545           |
> > > |                        | Random    | 0.528         | 0.544           |
> > > |                        | Adult     | 0.513         | 0.553           |
> > >
> > >
> > > **The performance on video retrieval (Nearest Neighbor Retrieval): Training in developmental order leads to better performance.**
> > > Following an excellent suggestion from the reviewer, to further evaluate unsupervised representation learning, we conducted nearest neighbor retrieval analysis. We followed the procedure described in Chen et al. 2021 and Duan et al. 2022. We conducted the evaluation on SSv2 and UCF-101 datasets and found that training in developmental order led to better performance in all cases, which is consistent with our overall conclusions about the benefits of the developmental curriculum. The results are provided in the table below.
> > >
> > > | Stage                  | Curriculum | SSv2 Top-1 | SSv2 Top-5 | UCF101 Top-1 | UCF101 Top-5 |
> > > |------------------------|------------|------------|------------|--------------|--------------|
> > > | 1 (After Iteration 30k)| Dev.      | **0.217**      | **0.579**      | **0.263**        | **0.393**        |
> > > |                        | Adev.     | 0.117      | 0.427      | 0.133        | 0.233        |
> > > |                        | Random    | 0.191      | 0.530      | 0.204        | 0.329        |
> > > |                        | Adult     | 0.190      | 0.535      | 0.220        | 0.348        |
> > > | 2 (After Iteration 60k)| Dev.      | **0.254**      | **0.625**      | **0.265**        | **0.428**        |
> > > |                        | Adev.     | 0.224      | 0.602      | 0.247        | 0.391        |
> > > |                        | Random    | 0.235      | 0.594      | 0.264        | 0.415        |
> > > |                        | Adult     | 0.222      | 0.580      | 0.264        | 0.411        |
> > > | 3 (After Iteration 90k)| Dev.      | **0.293**      | **0.657**      | **0.289**        | **0.451**        |
> > > |                        | Adev.     | 0.241      | 0.609      | 0.269        | 0.420        |
> > > |                        | Random    | 0.256      | 0.619      | 0.263        | 0.424        |
> > > |                        | Adult     | 0.245      | 0.610      | 0.269        | 0.424        |
> > >
> > >
> > > Chen, P., Huang, D., He, D., Long, X., Zeng, R., Wen, S., ... & Gan, C. (2021, May). Rspnet: Relative speed perception for unsupervised video representation learning. In Proceedings of the AAAI Conference on Artificial Intelligence (Vol. 35, No. 2, pp. 1045-1053).
> > >
> > > Duan, H., Zhao, N., Chen, K., & Lin, D. (2022). Transrank: Self-supervised video representation learning via ranking-based transformation recognition. In Proceedings of the IEEE/CVF Conference on Computer Vision and Pattern Recognition (pp. 3000-3010).
> > >
> > > We thank you again for your valuable feedback!

---

> > > > ### Comment · Reviewer_Bo4Q · 2023-08-20
> > > >
> > > > Thanks for providing results of training on a larger number of interactions, which further support the overall conclusions regarding the benefits of the developmental curriculum. I have raised my score.

---

### Official Review · Reviewer_wvZP · 2023-07-05

**Soundness:** 4 excellent
**Presentation:** 4 excellent
**Contribution:** 4 excellent
**Rating:** 7
**Confidence:** 4

**Summary:**

The authors aim to explore whether the natural structures and regularities in infant visual experience acquired via video recording when infants wearing head-mounted cameras can facilitate pre-training self-supervised learning representation learners. The authors divide the infant data into 3 groups (according age group) and evaluate the importance of training with a curriculum aligned with developmental order. The other 3 curriculums are designed for comparisons. The experimental results prove the order of the curriculum matter via self-supervised learning tasks and downstream tasks.

**Strengths:**

1. Overall the paper is well-written and is easy to understand the goal of the paper.
2. In the community, we all know that data is essential. However, we pay less attention to the order of curriculum that infants naturally created for training visual systems. The motivation of the paper is to explore whether the order of the curriculum is essential for visual systems. The authors leverage data curated from a larger corpus of 500 hours of head camera videos collected by infants from 2 to 24 months of age for experiments. They propose a clear experimental protocol and training curriculums for comparisons. In the experiments, they show deep neural networks (i.e., VideoMAE) trained on developmentally-ordered datasets outperformed those trained on “developmentally incorrect” datasets. The discovery of the experiments is valuable to the machine learning community to revisit our training dataset and learning curriculums.
3. The pre-trained models are finetuned for downstream tasks. Again, the results show a consistent trend that those trained on developmentally-ordered datasets outperformed those trained on “developmentally incorrect” datasets. This indicates that the order in which humans generate data for learning may therefore be an important contributor to achieving adult-like visual processing. This again provides the community to rethink our training protocols.
4. The reviewer particularly enjoys reading section 5.3. Specifically, the authors propose a methodology to probe the learned visual features using the developmentally-ordered datasets to provide a “peak” of infants’ receptive field. The findings could provide a new tool for developmental psychology researchers to design new experiments to validate the effectiveness of the methodology.

**Weaknesses:**

Overall, the reviewer enjoys reading the manuscript and learns a ton. I have the following questions, comments, and suggestions:

1. In Sec. 5.3, the authors highlight that it takes 12 components of g0 to explain 95% of the variance; whereas, it takes 33 components for g2. Overall, the larger number for g2 makes sense. The reviewer is surprised by the difference. The reviewer would like to learn from the authors whether the ratio of the two “number of components” could be some indicator. Assuming there is one, the ratio could potentially be used to validate the effectiveness of the learning models.

2. The main “concern” of the work is the paragraph in L287 to L300. Indeed, ViT has made tremendous progress in the vision community. The authors further utilize it on the developmentally-ordered dataset. However, as pointed out in the limitation section, there are other learning systems that could be explored. The reviewer is concerning the masking used in VideoMAE is the appropriate learner, while showing promising results in the work. In addition, the recent LLMs show promising results of autoregressive models (prediction of the next frame) compared to BERT-like approaches (filling masked areas). Moreover, self-attention is one of the important components. The reviewer brings these points to highlight the need to explore other existing learners so that we can provide a more thorough perspective to support the statements in L287 to L300.

3. Implementation details, Training protocol: while the experimental results do empirically prove the order of curriculum is essential, I wonder how the authors determine the number of iterations. In the experiments, 5000 iterations are used. Does the varying number of iterations influence self-supervised pretraining? Does the varying number of iterations influence finetuning downstream tasks?

**Questions:**

The questions are listed in the Weakness section.

**Limitations:**

Yes.

---

> ### Author Rebuttal · Authors · 2023-08-09
>
> We thank the Reviewer for detailed comments and for pointing out a number of strengths. Below we addressed each question mentioned by the Reviewer.
>
> **Reviewer** asked whether the ratio of the two “number of components” could potentially be used to validate the effectiveness of the learning models.
>
> **Our response:** As the Reviewer pointed out, the ratio of the number of principal components needed to explain 95% of the variance in g0 and g2 was indeed large. This reflects the dramatic difference in the infant visual inputs across development. We agree with the Reviewer that the ratio could potentially be used to characterize and compare datasets in terms of their similarity to the human visual inputs.
>
> **Reviewer** suggested to explore other existing learners and mentioned prediction of the next frame as a possible learning objective.
>
> **Our response:** We agree that this is important and we conducted additional evaluations using JEPA-TT and SimCLR-TT (see general response for details and Fig. R2 in the rebuttal PDF). Unlike VideoMAE, which is trained on reconstruction (similar to BERT-like approaches also mentioned by the Reviewer), JEPA-TT is trained on prediction using contrastive self-supervised learning through time. The results from JEPA-TT were qualitatively similar to the results from VideoMAE. SimCLR-TT is another approach based on contrastive learning (Aubret et al. 2022), which uses CNN instead of transformer architecture. We found again that the results remained qualitatively similar, supporting our general claim about the importance of the order of training data.
>
> **Reviewer** asked whether the varying number of iterations influence self-supervised pretraining and finetuning downstream tasks.
>
> **Our response:** The choice of 5000 iterations for each of the three stages of training was inspired by the Neyshabur et al. study, which reported the effect of curriculum going away if the number of iterations reached around 20k. Fig. S3 in the manuscript shows the results of the pretraining (panel A) and downstream classification (panel B) for 2000, 5000 and 15000 interactions per stage. Even when training for 15000 interactions, the anti-developmental curriculum did not result in improvement after the first stage (data from the oldest group, g2). Training for 15000 interactions in the anti-developmental curriculum also showed that the last stage (data from the youngest group, g0) made significant progress, but the results of the developmental curriculum were still better than those of the anti-developmental curriculum.
>
> We would like to add that comparing learning in humans and machines remains methodologically challenging. While there is strong evidence that sensorimotor learning has a large statistical learning component, potentially making use of error gradients, the details of how statistical visual learning is implemented in the brain is largely unknown. Nevertheless, the recent powerful deep learning algorithms provide a method to ask broad questions about the relationship between natural data streams and the learning outcomes, behavioral or neural. Here, our goal was to ask how the properties of the real world experience of infants affect general statistical learners. Along the way, we have done our best to choose the learning system with the goal of minimizing the role of factors that make it less humanlike (e.g., very large batch sizes, many epochs, etc) while still benefiting from the scalability of deep learning systems in learning from relatively large scale real-world data.

---

> > ### Comment · Reviewer_wvZP · 2023-08-18
> > **Thank you for the response!**
> >
> > Dear Authors,
> > Thank you for the detailed response! The response addresses my concerns, particularly on using additional evaluations using JEPA-TT and SimCLR-TT!
> >
> > After reading the results, I have the following questions:
> > 1. Do the finding found in Sec. 5.3 still holds for JEPA-TT and SimCLR-TT?
> > 2. The Neyshabur et al. study (which reported the effect of curriculum going away if the number of iterations reached around 20k) should be cited in the manuscript. While using the evidence as a starter for experiments makes sense, the reviewer found it would make the paper stronger if the authors conduct thorough experiments to analyze if the statement in Neyshabur still holds in using the videos.

---

> > > ### Author Response · Authors · 2023-08-18
> > >
> > > Thank you for the comment and the questions. Please see our answers below.
> > >
> > > 1. Yes. The PCA analysis is performed directly on the videos, regardless of the learning algorithm.
> > >
> > > 2. Our ablation studies include a condition (Fig S3) in which we pretrained our models on 6k iterations (2k iterations x 3 stages), 15k iterations (5k iterations x 3 stages) and 45k iterations (15k iterations x 3 stages). We observed that the effect of the curriculum reduces but does not completely go away. We note that Wu et al. 2020 study (which we mistakenly labeled by the last author Neyshabur in the rebuttal) investigates curriculum effects in supervised learning with manually designed curricula, which is somewhat different from our self-supervised learning in a natural curriculum setting. Therefore, some difference in the size of the curriculum effect is expected. Wu et al. 2020 study is cited in line 264 of the manuscript (we apologize for mentioning it as Neyshabur et al. in the rebuttal).
> > >
> > > Xiaoxia Wu, Ethan Dyer, and Behnam Neyshabur. When do curricula work? arXiv preprint arXiv:2012.03107, 2020.

---

> > > > ### Comment · Reviewer_wvZP · 2023-08-19
> > > > **Thank you for the response**
> > > >
> > > > Dear authors,
> > > > Thanks for your response! My concerns are addressed. Please incorporate the additional details in your final version.
> > > > Thanks for sharing your work, and I learn a lot from it!

---

> > > > > ### Author Response · Authors · 2023-08-19
> > > > >
> > > > > Thank you, we very much appreciate your comments and feedback. We will incorporate the additional details into the final version of the manuscript.

---

### Official Review · Reviewer_KVqq · 2023-07-06

**Soundness:** 3 good
**Presentation:** 2 fair
**Contribution:** 3 good
**Rating:** 6
**Confidence:** 3

**Summary:**

The work is essentially about an experimental evaluation to show the benefit of curriculum learning using ego-centric videos acquired with from head-mounted cameras and very young children. The authors discuss the use of self-supervised learning on the data ordered according to developmental principles and empirically show the improvement that can be observed on down-stream tasks and benchmark datasets.

**Strengths:**

I find the idea of coupling a developmental approach with curriculum learning very fascinating, and deserving attention.
The main motivations behind the developmental paradigm that inspired the approach are clearly reported.

**Weaknesses:**

I have some main concerns about the clarity of the presentation and the contributions that make me think the work is not yet ready for publication at this conference:

- On the motivations behind the developmental approach: I find it fascinating, but somehow in contrast with the use of large data sets and complex architectures. On one side there are children with their visual, cognitive and motor abilities under development, on the other these very complex architectures. For the latter, I think a more convincing justification should be provided on why those have been used.

- While the overall inspiration and philosophy of the work are clear (although sometimes maybe not immediately clear for those with no particular knowledge of Cognitive Science) in general, I found a lack in the justifications of the more technical choices. Just as a possible example: from row 157 to row 165, more details in the questions.

- Since the main objective was to show the benefit of the developmental approach, my impression is that the limitation correctly highlighted by the authors of having considered just one architecture is too important to be disregarded. In my opinion, showing that the principles observed in these experiments emerge also when changing the architecture is fundamental to speak in favour of the generality of the results

**Questions:**

I will try to list some more specific questions that I hope will help to clarify some important points:

- Here and there there are somehow strong statements that would require appropriate citations. For instance: "A common assumption in the machine-learning literature is that at the massive scale of daily life, the world presents the same visual statistics to all perceivers. Many researchers therefore believe that it will be possible to mimic biological learning through incremental statistical learning performed  over large datasets.", or also  “An appropriate model and the objective function to assess a developmental curriculum is one with a  self-supervised learning objective  and model architecture that directly learns the regularities in the video dataset across both space and time – a spatiotemporal learner. “

- Has the dataset been acquired by the authors? If yes, why isn’t it a contribution of the work? On the basis of what the videos have been ordered on the basis of the motor abilities? In the paper it is said: “…we selected videos collected by infants at 3 age periods who differ in the motor skills and by hypothesis in the statistics of experience” Who made the selection? I think specific expertise would be needed

- Again on the dataset: In the dataset description is n referring to the number of children or videos? If these are the children, a significant corpus of acquisitions has been discarded from the original 101 to the 29 that have been used. Why? Wouldn’t it be possible to use them as an extra, maybe more complex, test set?

- The two training sets (children and adults) are very imbalanced: 123h for children and 64 for adults. This can negatively affect the results and would deserve some comments

- Row 173: why only edge features?

- Row 187: why a linear classifier?

- Motivations behind the use of video-masked autoencoders?

- Sect. 5.1: is the validation loss computed on a validation set of the same class of data? E.g. is the loss for the model trained on g0 computed on validation from g0? If yes, this may explain the different performance (g0 simpler than others?). If not, it must be more clearly explained

- “…we first tracked the self-supervised loss of the VideoMAE model trained separately with the data from the youngest age group (g0), the data from the oldest age group (g2), random group, and adult group” This is not what we see in Figure 2, where in the legend I see instead, for instance, “Dev. (g0-g1-g2)”. How should we relate the text with the figure? It is not clear to me, I failed to track the results. From what I understand Figure 2 is pertinent only to Sec, 5.2 (maybe also the experiments from 5.1 are there, but too hidden to the appreciated)

- I’m not sure I understood what validation loss is used in Sec. 5.1: is it based on reconstruction? If yes, why it starts from 1?

- “ Our results also provide evidence in favor of the long-standing perspective in  neuroscience that emphasizes the importance of slowly changing sensory signals in visual learning for the emergence of sparse and efficient coding. “ I fail to get this observation from your results, where should it be emerging?

**Limitations:**

The limitations of the work have been mentioned in a specific section

---

> ### Author Rebuttal · Authors · 2023-08-09
>
> We thank the Reviewer for detailed comments and for the enthusiasm about our approach. Below are our responses to the Reviewer’s questions.
>
> **Reviewer** asked for a more convincing justification on why we used large data sets and complex architectures.
>
> **Our response:** While we did use advanced architectures, the number of neurons in those architectures is still a fraction of those in a newborn's brain (estimated to be around 100 billion).
>
> **Reviewer** found a lack in the justifications of the technical choices.
>
> **Our response:** With the proposed additions (rebuttal PDF), we are covering state-of-the-art self-supervised architectures and loss functions. In general, we chose self-supervised learning to reflect the nature of learning by human infants.
>
> **Reviewer** stated that ‘’showing that the principles observed in these experiments emerge also when changing the architecture is fundamental to speak in favour of the generality of the results’’.
>
> **Our response:** We agree with this point and we conducted an evaluation using JEPA-TT and SimCLR-TT. For more details, please see general response and Fig. R2 in the rebuttal PDF.
>
> **Reviewer** asked to support with more citations some of the general claims made in the manuscript and pointed out two such statements.
>
> **Our response:** We agree with the Reviewer and we plan to add the following references to support the first statement: Torralba et al. 2003, Felsen et al. 2005, Simoncelli et al. 2001, Lake et al. 2017 and following references to support the second statement: Zaadnoordijk et al. 2022, Orhan et al. 2020, Zhuang et al. 2019.
>
> **Reviewer** asked whether the dataset has been acquired by the authors.
>
> **Our response:** We used the Homeview human-subjects dataset, which is not publicly available due privacy, rights and confidentiality of the participants.
>
> **Reviewer** asked about the basis for ordering of the videos into 3 groups.
>
> **Our response:** The first 12 months are critical for proper development of human visual systems and we choose 3 stages within that period that are characterized by very different properties of the visual inputs. Very young infants, for example, prefer to look at scenes with large, simple high contrast edges, and they are “sticky” lookers [Smith et al. 2018], orienting to and looking at a single object for many seconds. In the following months, there are marked and systematic increases in the control, complexity, purposefulness, speed and precision of body movements that include rapid and more varied changes in head and body orientation, reaching for, handling and performing instrumental acts on objects and self-locomotion through crawling and walking [Kretch et al. 2014]. We provided more details in lines 120-129 in the manuscript.
>
> **Reviewer** asked why not all recorded data from the Homeview dataset was used.
>
> **Our response:** We focused on three stages during the first 12 months and we did not use data from participants outside of those stages (many participants were older than 12 months).
>
> **Reviewer** said that two training sets (children and adults) are very imbalanced: 123h for children and 64 for adults.
>
> **Our response:** To compensate for the imbalance, we bootstrapped the adult data to match the size of the children data.
>
> **Reviewer** asked why we intuit each token in VideoMAE as a small, moving edge feature.
>
> **Our response:** The sentence was intended to provide an intuition to those less familiar with transformer architecture. We understand it might cause confusion, so we will remove the moving edge intuition.
>
> **Reviewer** asked why we used a linear classifier.
>
> **Our response:** Downstream classification using a linear classifier has been a common practice for evaluating self-supervised learning algorithms in recent years [Orhan et al. 2020, Aubret et al. 2022, Assran et al. 2023, Chen et al. 2020].
>
> **Reviewer** asked about motivations for using video-masked autoencoders.
>
> **Our response:** They are self-supervised spatiotemporal learners with relatively few design choices and hyperparameters.
>
> **Reviewer** asked if the validation loss computed on a validation set of the same class of data and whether that might explain different performance.
>
> **Our response:** A separate validation set was constructed for each age group. Since we observed lower loss for developmental curriculum after each of the three stages, the overall result does not appear to be determined by the properties of the validation set. Furthermore, performance on the downstream classification tasks was independent of the validation set and it was consistently better for the developmental curriculum.
>
> **Reviewer** asked about visualization of the results from Stage 1 shown in Fig. 2.
>
> **Our response:** The beginning of each stage is marked with a dashed line, an arrow and a label. Each stage has 5000 iterations, so the results for Stage 1 are at 5000-th iterations. To avoid confusion, we will replace the visual language in the current figures with the visual language shown in the rebuttal PDF (bar plots).
>
> **Reviewer** asked whether the validation loss in Sec. 5.1 is based on reconstruction and why it starts from 1.
>
> **Our response:** Validation loss is based on reconstruction and was normalized to always start at 1.
>
> **Reviewer** asked about the support for the sentence about slowly changing sensory signals.
>
> **Our response:** Indeed, the difference in learning between developmental and anti-developmental curricula could have emerged due to spatial and/or temporal differences in the data. To better understand the causes of the difference, we performed new analyses in which we controlled for spatial and temporal complexity (see general response for details). As shown in Fig. R1 (rebuttal PDF), temporal complexity was particularly important for driving the difference between developmental and anti-developmental curricula.

---

> > ### Comment · Reviewer_KVqq · 2023-08-18
> > **Thank you for the response!**
> >
> > I thank the authors for their very detailed response, my main concerns have been addressed.Although I think there might be room for improvements, I raised my rating since I find the approach interesting and deserving attention

---

> > > ### Author Response · Authors · 2023-08-19
> > >
> > > We thank the Reviewer for the feedback and for raising the score.

---

### Official Review · Reviewer_L6Nk · 2023-07-07

**Soundness:** 3 good
**Presentation:** 2 fair
**Contribution:** 3 good
**Rating:** 6
**Confidence:** 4

**Summary:**

This paper aims to study how developmental changes impact visual learning. It used infant egocentric videos to pre-train video autoencoders with self-supervised learning loss. It separated the infant data by age group and evaluated the importance of training with a curriculum aligned with developmental order.

**Strengths:**

The study goal of this paper is interesting. Few has studied the developmental changes in infant visual experience and characterize how those changes impact visual development.

**Weaknesses:**

1) The contribution of this paper is limited. The datasets are collected from existing datasets. Only single existing video autoencoder model is used and evaluated. While the goal of the paper is interesting, it is not conducted in a professional way.
2) The writing and organization of the paper need much improvement. It is not easy to follow the content of the paper.

**Questions:**

None.

**Limitations:**

Limitations of the paper and future studies are discussed.

---

> ### Author Rebuttal · Authors · 2023-08-09
>
> We thank the Reviewer for the comments and appreciate the Reviewer's enthusiasm about the goals of the paper. Below we addressed each weakness mentioned by the Reviewer.
>
> **Reviewer:** The contribution of this paper is limited.
>
> **Our response:** We believe that the manuscript delivers several substantial contributions:
> We demonstrated that not only do data sets matter, the order of data sets matters.
>
> Evolution has taken great pains to control the order of visual experiences for infants, and our results showcase that AI researchers might benefit by providing machines with a common, naturally-ordered learning curriculum.
>
> Our results indicate that machines learn best when they first receive training data that changes very slowly, with a bias to input characterized by high contrast and large simple edges. This “edge-based” learning (V1 in mammalian cortex) is the first step in the formation of high-level visual representations.
>
> The order of the data matters, regardless of the kind of learner algorithm. This conclusion is further supported by the experiments discussed in the rebuttal. This indicates that our results reflect a general principle about learning.
>
> **Reviewer:** The datasets are collected from existing datasets.
>
> **Our response:** The dataset used here is the only one of its kind ever recorded. It captures the ordered experiences of human infants in the first year of life, a period of marked learning and tuning of the human visual cortex. These are daylong recordings, all daily-life contexts, no experimenters present who might distort the statistics. This manuscript is the first time that the dataset has been used to train artificial neural networks and thus we believe it provides a valuable insight into similarities and differences of human and machine learning.
>
> **Reviewer:** Only a single existing video autoencoder model is used and evaluated.
>
> **Our response:** We conducted additional evaluations using JEPA-TT and SimCLR-TT (see general response for details and Fig. R2 in the rebuttal PDF). Unlike VideoMAE, which is trained on reconstruction, JEPA-TT is trained on prediction using contrastive self-supervised learning through time. The results from JEPA-TT were qualitatively similar to the results from VideoMAE. SimCLR-TT is another approach based on contrastive learning (Aubret et al. 2022), which uses CNN instead of transformer architecture. We found again that the results remained qualitatively similar, supporting our general claim about the importance of the order of training data.
>
> We also introduced an additional indicator of the learning quality, the gradient norm. This metric provided an insight into the learning process and clearly separated developmental and anti-developmental curricula (see general response for details and Fig. R1A, Fig. R2A and Fig. R2B in the rebuttal PDF).
>
> Furthermore, to gain deeper insight into why learning is better when data comes in developmental order, we conducted an analysis where we controlled for spatial and temporal complexity. This revealed the importance of the temporal properties of the training data, suggesting that training with slower data first will improve learning (see general response for details and Fig. R1 in the rebuttal PDF).
>
> **Reviewer:** The writing and organization of the paper need much improvement.
>
> **Our response:** We will do our best to edit the manuscript to improve its readability. We already changed the visual language in the attached figures in the rebuttal PDF and we plan to apply that visual language to other figures in the manuscript. We welcome any suggestion from the Reviewer regarding the organization of the manuscript.

---

> > ### Comment · Reviewer_L6Nk · 2023-08-21
> >
> > Thanks for the detailed responses. Most of my initial concerns are addressed with clear explanation of the contributions and additional experiments. I would raise my rating. Please further refine the manuscript into a more readable version.

---

### Author Rebuttal · Authors · 2023-08-09

Our paper tackles a question at the heart of human and machine learning: How do we compare the learning abilities of human infants and machines? Our study takes an important step in this direction by (1) showing that generic learning algorithms (with no hardcoded knowledge about objects or space) can learn to solve complex real-world vision tasks when the models are trained “through the eyes” of human infants; (2) showing that the order of training data matters, such that machines learn best when they receive training data in the same order as human infants; and (3) showing that slow visual inputs are valuable for kick-starting visual learning. The dominant assumption in AI is that the order of training data does not matter for large-scale computer vision systems, so our results contradict a core assumption in the field. Our study also introduces the first computer vision system that has been trained in a “developmentally correct” order, laying a critical foundation for reverse engineering the learning algorithms in human brains.

The Reviewers largely agreed with our goals and approach. However, the ratings were ambivalent for four reasons:

First, in the original submission, we only tested one ViT model (VideoMAE). To address this critique, we added new experiments testing JEPA-TT models, which learn via contrastive learning through time. As shown in the left column in Fig. R2 (rebuttal PDF), the JEPA-TT model showed similar learning patterns as the VideoMAE model. This result indicates that our conclusions reflect general principles of learning across biological and artificial systems.

Second, in the original submission, we did not compare ViTs to CNNs. It was thus unclear whether the spatial inductive bias of CNNs helps or hinders performance. To address this critique, we added new experiments testing CNNs that learn via contrastive learning through time (SimCLR-TT; Aubret et al. 2022). As shown in the right column in Fig R2 (rebuttal PDF), both CNNs and ViTs were sensitive to the order of the training data. This finding shows that the learning effects we reported in our original paper generalize across different architectures (ViTs and CNNs).

Third, the Reviewers worried that since we did not provide causal evidence that slow visual inputs drive rapid learning in ViTs, our conclusions were limited in scope. To address this critique, we performed new analyses in which we controlled for spatial and temporal complexity. As shown in Fig. R1 (rebuttal PDF), the learning differences between models trained on developmentally accurate vs. developmentally inaccurate data largely disappeared. These analyses show that slow visual inputs are causally connected in the rapid development of machine vision, akin to slow visual inputs being causally connected to the development of biological vision.

Finally, Reviewers questioned the scope of the project and importance of the results. To address these critiques, we will revise our manuscript to emphasize that our study is the first of its kind, providing a unique opportunity to directly compare the learning abilities of human infants and computer vision models. We would also emphasize that our results have important scientific implications. ViTs have the potential to be powerful image-computable models of human visual systems, but these models will not be accepted if they are thought to learn differently from humans. Our results contradict this assumption, paving the way for high-performing models from AI to serve as image-computable models of visual learning and development.

Below we provide details for the methods used to generate the figures in the rebuttal PDF.

JEPA-TT

I-JEPA (Assran et al., 2023) is a recently proposed Image-based SSL algorithm in which the objective is to predict the embedding of one part of the input image (target) using the embedding of another part (context). Our JEPA Through Time (JEPA-TT) algorithm is an adaptation of I-JEPA where the context and the target are selected from consecutive video frames rather than the same frame.

SimCLR-TT

SimCLR is a well-known Image-based SSL algorithm. It works by bringing the embeddings of two augmented versions of an image (positive pair) closer in the embedding space while pushing them away from embeddings of randomly selected images from the dataset. Similar to Aubret et al. 2022, our SimCLR-TT algorithm treats the consecutive frames as augmentations of each other. However, we found this method to overfit our dataset very quickly, therefore we reduced the number of training iterations from 5000 to 1500 (analogous to early stopping).

Controlling for temporal and spatial complexity

To match temporal complexity, we used individual video frames as the model input, instead of a series of frames. To match spatial complexity, we equalized the number of edges (as a measure of image complexity) in images from each of the three age groups. To achieve this, we first extracted the edges using a Canny edge detector and then sampled images to match the number of edge pixels. We found that the gap in the metrics between the developmental and anti-developmental curricula reduced as we matched the spatial complexity, temporal complexity, or both (Fig. R1).

Gradient Norm

We report an additional indicator of the learning process, the gradient norm, computed as $||\frac{1}{batch\ size} \sum \frac{\partial \text{Loss}}{\partial \Theta}||_2$ where $\Theta$ contains the learnable weights of a given layer. This quantity has been used for studying the learning dynamics of Stochastic Gradient Descent (SGD) (Shwartz-Ziv and Tishby, 2017). We found that regardless of the learning objective and architecture, the developmental curriculum had the largest gradient norm in the first stage of training, where it receives the data stream from the youngest infants (g0). Accordingly, the anti-developmental curriculum receives the g0 data at the third stage and has the larger gradient norm.

---

### Decision · Program_Chairs · 2023-09-21

**Decision:**

Accept (spotlight)

**Comment:**

The paper studied the curriculum learning of visual representation based on inputs gathered from infant's point of view. While the study considered existing techniques in self-supervised learning, the results demonstrated some some intriguing findings about the importance of training with a curriculum aligned with developmental order.

The initial review ratings were mixed. The reviewers raised concerns on the experiments, and questioned some of the technical details. The authors provided a detailed response that addressed most of the concerns. After the rebuttal, the reviewers applaud the findings and contributions of the paper. The decision is thus to recommend its acceptance.